# Genetic Variation in Common Bunt Resistance in Synthetic Hexaploid Wheat

**DOI:** 10.3390/plants12010002

**Published:** 2022-12-20

**Authors:** Amira M. I. Mourad, Alexey Morgounov, P. Stephen Baenziger, Samar M. Esmail

**Affiliations:** 1Leibniz Institute of Plant Genetics and Crop Plant Research (IPK), 06466 Gatersleben, Germany; 2Agronomy Department, Faculty of Agriculture, Assiut University, Asyut 71526, Egypt; 3International Maize and Wheat Improvement Center (CIMMYT), P.K. 39 Emek, 06511 Ankara, Turkey; 4Department of Agronomy and Horticulture, Plant Science Hall, University of Nebraska–Lincoln, Lincoln, NE 68583, USA; 5Wheat Disease Research Department, Plant Pathology Research Institute, Agricultural Research Center, Giza 12619, Egypt

**Keywords:** *Tilletia caries*, *T. Foetida*, correlation, differential lines, seedling vigor

## Abstract

Common bunt (caused by *Tilletia caries* and *T. Foetida*) is a major wheat disease. It occurs frequently in the USA and Turkey and damages grain yield and quality. Seed treatment with fungicides is an effective method to control this disease. However, using fungicides in organic and low-income fields is forbidden, and planting resistant cultivars are preferred. Due to the highly effective use of fungicides, little effort has been put into breeding resistant genotypes. In addition, the genetic diversity for this trait is low in modern wheat germplasm. Synthetic wheat genotypes were reported as an effective source to increase the diversity in wheat germplasm. Therefore, a set of 25 synthetics that are resistant to the Turkish common bunt race were evaluated against the Nebraska common bunt race. Four genotypes were found to be very resistant to Nebraska’s common bunt race. Using differential lines, four isolines carrying genes, *Bt10*, *Bt11*, *Bt12*, and *Btp*, were found to provide resistance against both Turkish and Nebraska common bunt races. Genotypes carrying any or all of these four genes could be used as a source of resistance in both countries. No correlation was found between common bunt resistance and some agronomic traits, which suggests that common bunt resistance is an independent trait.

## 1. Introduction

Common bunt caused by *Tilletia caries* (D.C.) Tul (=*T. tritici*) and *T. foetida* (Wall.) Liro (=*T. laevis*) occurs frequently in most of the global wheat-growing areas and causes huge losses in wheat yield and quality [1]. In the infected plants, kernels are fully filled with bunt balls that are full of spore masses of the common bunt [2]. In addition, the harvested healthy grain is rejected by millers due to its noticeable odor in the wheat kernel and flour [3].

At the beginning of the 20th century, chemical fungicide treatment was applied to seeds to control common bunt infections. Due to the effectiveness of these fungicide treatments, research on the genetics of common bunt resistance and resistant genotypes has generally been neglected. For this reason, there is a low number of highly resistant commercial cultivars currently available [4,5,6,7]. The disease has recently re-emerged in the Great Plains area of the USA and in Turkey, especially in organic fields where fungicides are forbidden [5,8,9,10]. In the developing world where seed treatments are expensive or not available, the common bunt is also a concern [2]. For organic and low-input producers, resistant cultivars are preferred.

Unfortunately, the genetic diversity in modern germplasm is limited for this trait, which hinders the selection of common bunt-resistant genotypes [11,12]. Therefore, there is a need to increase the genetic diversity in wheat to be able to breed common bunt-resistant genotypes. Hexaploid synthetic hybrid wheat genotypes derived from crosses between durum wheat (*T. durum*) containing genomes A and B with *Aegilops tauschii* containing genome D is an important source of genetic diversity of useful wheat traits as well as resistance to fungal diseases [13,14]. At the International Maize and Wheat Improvement Center (CIMMYT), winter synthetic wheat genotypes development began in 2004 by crossing winter durum wheat germplasm from Ukraine and Romania with winter *Ae. tauschii* accessions. The pedigree selection was applied to the F_3_ populations of these crosses using drought resistance, cold resistance, and disease resistance as selection criteria from 2009 to 2015 [15]. Superior synthetic hybrid wheat genotypes which were resistant to common bunt could be used as a source for common bunt resistance in Turkey and other countries such as the United States after evaluating them with the common bunt race that exists in the targeted country.

The first step in producing genotypes that are resistant to different common bunt races in different countries is to identify the resistant genes to each race separately and for both races. Common bunt resistance genes are expressed as *Bt* genes. Until now, there have been sixteen resistance genes from *Bt1*–*Bt15* in addition to *Btp* [16]. To identify the resistance genes in a specific common bunt race/strain, the worldwide common bunt differential set should be used [17]. This differential set contains isolines for each resistant gene which differ in their growth habit from spring and winter wheat. After the determination of resistant genes against the targeted race/s, a DNA-based marker should be used to confirm the presence of the identified genes in the resistant genotypes. Some of the resistance genes have been chromosomally mapped in the wheat genome [18,19,20,21]. However, there are no available markers for all the resistance genes and the available markers are still preliminary as their accuracy is questionable [4,22,23]. In recent times, more concern has been given to identifying qualified DNA markers by identifying the genomic regions controlling the resistance using a genome-wide association study (GWAS) and quantitative trait loci (QTL), which are mapped to transfer them into different types of markers such as a Kompetitive allele-specific PCR (KASP) [12,24,25,26,27,28,29]. However, it will take a long time to achieve this target. Due to these complications, the only way to improve common bunt resistance is to preliminarily predict the resistant genes using differential lines and cross the resistant genotypes to increase the genetic diversity for common bunt resistance.

The objectives of this study are to (1) screen a set of 25 elite synthetic wheat genotypes to common bunt resistance under Nebraska conditions to identify the resistant genotypes that can be used as a possible source of resistance genes in Nebraska breeding programs, (2) identify possible genes that resist both Nebraska and Turkish common bunt races, and (3) study the correlation between common bunt resistance and some agronomic traits.

## 2. Materials and Methods

### 2.1. Plant Materials

A total of 41 genotypes were used in this study. These genotypes could be classified into two main groups: common bunt isolines (differential lines) and tested genotypes. The common bunt isolines consisted of 14 differential lines which were discussed previously in Amira M. I. Mourad et al. (2018). In brief, a set of 12 lines were winter genotypes (from *Bt1* to *Bt13* (*Bt4* and *Bt5* were excluded due to the lack of seeds)). Two genotypes, *Bt14* and *Bt15*, were spring wheat. In addition, two common bunt-susceptible genotypes, ‘Red Bobs’ and ‘Heines VII’, were included in this study as checks to confirm the success of the artificial inoculation (Table 1).

The tested genotypes could be classified into three groups: checks, CIMMYT winter wheat synthetics, and lines originating from the CIMMYT winter synthetics crossed to Turkish cultivars (Table 2). The two check genotypes were winter wheat genotypes, which are adapted to the Turkish environment and are usually planted in rainfed fields. The second group consisted of some winter synthetic hybrid wheat genotypes produced by the CIMMYT by crossing winter durum wheat cultivars from Ukraine and Romania to *Ae. tauschii* winter accessions, and they were selected based on agronomic performance and disease resistance [15]. The third group was produced by crossing some of the winter synthetic genotypes produced by the CIMMYT with modern Turkish cultivars.

### 2.2. Common Bunt Inoculation

The seeds of the tested genotypes, as well as the differential lines, were inoculated with common bunt spores by putting them in an envelope with the teliospores and shaking them well until the seeds were fully covered with the teliospores. This method was reported as an effective one to inoculate a small number of seeds artificially (from five to twenty grams) [3,28].

### 2.3. Experimental Design

The tested genotypes and the 12 winter wheat differential lines were tested for their common bunt resistance in a field at two locations, Lincoln and Mead, Nebraska, USA, in the 2015/2016 season. In addition, the susceptible winter check “Heines VII” was included in the field experiment in both locations to confirm the effectiveness of the artificial inoculation. All 38 entries (25 tested genotypes, 12 winter isolines, and Heines VII) were planted using a randomized complete block design (RCBD) experimental design with two replications in each of the two locations. The seeds of each genotype were sown in a one-meter-long row at a five cm depth. The soil temperature at this depth was 18 °C and 17 °C at Lincoln and Mead, respectively ((http://hprcc.unl.edu/) accessed on 1 February 2017).

The two spring differential lines “*Bt14*, and *Bt15*” as well as the spring susceptible check “Red Bob” were evaluated for common bunt resistance in the greenhouse. The experimental design was an RCBD with three replications. In each replication, four genotypes were planted in each pot. The twelve winter differential lines were included in the greenhouse experiment along with the spring differential lines to ensure that none of the differentials escaped from the infection under the field conditions. All the tested genotypes, spring and winter genotypes, were kept in the vernalizer after inoculation for two months at 4 °C with 12h light to provide optimal conditions for the fungal spores to infect the seedlings. After two months, the seedlings were transferred to a warm room at 16 °C/night and 25 °C/day till maturity [3].

For the tested genotypes, the following traits were scored: seedling vigor (was scored visually on 50% of the plants/line using a scale from 1 to 9, whereby “1” referred to weak seedlings and “9” referred to vigorous seedlings), days to heading (number of days after 1 January 2016 to when 50% of the tillers had emerged heads (Feekes stage 10.1)), chlorophyll content (average chlorophyll content from five flag leaves using SPAD-502 (KONICA MINOLTA, New York, NY, USA [30] (Feekes stages 10.5)), and plant height (was measured as the height of 50% of the plants from the soil surface to the tip of the head; awns were excluded when the plants were at Feekes stages 11 and growth had finished).

Common bunt resistance was scored for all the genotypes (the tested genotypes, fourteen differential lines, and the two susceptible checks) as the percentage of infected heads using the following formula:(1)$$CB=\frac{number\;of\;infected\;heads}{\mathrm{total}\;\mathrm{number}\;\mathrm{of}\;\mathrm{heads}\;\mathrm{per}\;\mathrm{genotype}}\;*100$$

Genotypes with zero % infected heads were considered very resistant to common bunt. Genotypes with 0.1–5.0% infected heads were considered resistant to common bunt. Those with a percentage of infected heads of 5.1–10%, 10.01–30%, 30.1–50%, and 50.01–100% were considered moderately resistant, moderately susceptible, susceptible, and very susceptible genotypes, respectively [31]. A Field book Android application was used to collect data for all the studied traits [32].

### 2.4. Statistical Analysis of All the Studied Traits

For all the studied traits, data on the tested genotypes from both locations, Lincoln and Mead, were combined. An analysis of variance (ANOVA) for the combined data on the different traits was performed using PLABSTAT software [33] using the following model:Y*_ijk_* = *µ* + l*_j_* + r*_k_* + g*_i_* + lg*_ik_* + e*_ijk_*(2)
where Y*_ijk_* is the observation of genotype *i* in replication *k* at location *j*; *µ* is the general mean; l*_j_*, r*_k_*, and g*_i_*, are the mean effect of the location, replication, and genotypes, respectively; lg*_ik_* is the interaction between the genotypes and locations; and e*_ijk_* is the error. The genotypes were assigned as fixed effects while replications and locations were assigned as random factors.

Broad-sense heritability was calculated using PLABSTAT using the following formula:(3)$$H^{2}=\frac{\sigma_{G}^{2}}{\left(\sigma_{G}^{2}+\frac{\sigma_{LG}^{2}}{l}+\frac{\sigma_{e}^{2}}{lr}\right)}$$
where $\sigma_{G}^{2}$, $\sigma_{LG}^{2}$, and $\sigma_{e}^{2}$ are the variance of lines and residuals. l and *r* are the number of locations and replications, respectively.

## 3. Results

### 3.1. Evaluating the Differential Lines and Susceptible Checks

The winter differential lines, as well as the winter check, were evaluated for their resistance to the common bunt in the field at two locations, Lincoln and Mead. Based on the average of the two locations, the winter check Heines VII had a low percentage of infected heads (14.4%, Table 1). The spring check, as well as the two spring differential lines, were evaluated in the greenhouse. The winter differential lines were also included in the greenhouse experiment to make sure that none of them escaped from the infection. Based on the average of the three replications, the spring check “Red Bob” had a high percentage of infected heads (73.35%, Table 1). Out of the fourteen tested differential lines, seven differential lines (*Bt6*, *Bt9*, *Bt11*, *Bt12*, *Bt13*, *Bt15*, and *Btp*) had zero% infected heads and were considered very resistant to the Nebraska common bunt race. The differential lines *Bt10* and *Bt7* had 1.2% and 3.8% of the infected heads, respectively, and were considered moderately resistant to the Nebraska common bunt race. The remaining five differential lines, *Bt1*, *Bt2*, *Bt3*, *Bt8*, and *Bt14*, had a percentage of infected heads more than 10% and were considered moderately susceptible to susceptible (Table 1).

### 3.2. Evaluation of the Winter Synthetic Wheat Genotypes

The ANOVA of the common bunt resistance revealed highly significant differences between the tested genotypes as well as a significant G×L interaction (Table 3). A highly significant correlation between the common bunt resistance in the two locations was found (*r* = 0.64) (Figure 1). Four genotypes were found to be very resistant to common bunt at Lincoln, and these genotypes were found to have the same response to common bunt at Mead. However, in addition to the four genotypes with zero% infected heads at both locations, four more genotypes were found to be very resistant at Mead and moderately resistant at Lincoln. At Lincoln, seven genotypes were resistant to common bunt with a percentage ranging from 0.1% to 5.0%, while five genotypes were in this category of resistance at Mead (Figure 2). Four and two genotypes were moderately resistant with a percentage of infected heads ranging from 5.1% to 10% at Lincoln and Mead, respectively. Ten genotypes were moderately susceptible (10.01–30%, infected heads) to common bunt at Lincoln. Finally, six and four genotypes (30.1–50%, infected heads) were moderately susceptible and susceptible at Mead, respectively. The broad-sense heritability for common bunt infection was very high at 0.86 based on the average of the two locations.

### 3.3. Estimation of Agronomic Traits under Common Bunt Infection

Seedling vigor, plant height, chlorophyll content, and heading date were evaluated under common bunt infection to identify the correlation between the resistance and agronomic traits in wheat. The ANOVA revealed highly significant differences between the genotypes for seedling vigor and plant height under infection (Table 3), while no differences were found between the genotypes for chlorophyll content and heading date (data not shown). No significant differences were found between the locations for plant height, while highly significant differences were found between locations for seedling vigor as well as a significant G×L interaction. The broad-sense heritability was 0.64 and 0.69 for seedling vigor and plant height, respectively. Seedlings of the resistant and susceptible genotypes were growing strongly at Lincoln with an average seedling vigor of 7.5, and they were weak at Mead with an average of 4 (Figure 3a,b). The plant height of the resistant and susceptible genotypes was almost the same with an average of 94 and 90 cm, respectively, based on the average of the two locations (Figure 3c). No significant correlation was found between the percentage of infected heads and seedling vigor or plant height (*r =* −0.01 for both traits).

## 4. Discussion

The seedlings were planted on 14 October 2015 in both locations. During that time, the soil temperature was 18 °C and 17 °C at Lincoln and Mead, respectively. Cool soil temperatures are very important for the common bunt fungus to grow and infect wheat seedlings. During the growing season (2015/2016), snow covered the soil for two months, from mid-November until mid-January. A long period of snow cover is very important for the fungus to produce a high level of infection [3]. To consider the evaluation of a common bunt study as a valid one, the susceptible check “Heinses VII” should have a percentage of infected heads exceeding 50% [34]. In our experiment, Heinses VII had a very low percentage of infected heads in both locations, 17.81% and 23.00% at Lincoln and Mead, respectively, with an average of 14.42%. Such a low percentage of infected heads could cause some suspicion about the results of our study. However, the evaluation of the tested genotypes was part of a study of common bunt resistance in Nebraska winter wheat germplasm, where very susceptible genotypes were found [28]. The low percentage of infected heads in Heines VII could be due to the fact that it was not adapted to Nebraska conditions. In addition, the absence of very susceptible genotypes among the tested synthetic wheat genotypes was expected as these genotypes were selected for common bunt resistance under Turkish conditions, so some resistance genes are expected to be in the genetics of these genotypes. Based on the existence of very susceptible genotypes in the Nebraska winter wheat germplasm [28], we can consider our evaluation trial as a valid one.

### 4.1. Genetic Variation in Common Bunt Resistance in Synthetic Hexaploid Wheat

Highly significant differences were found between the tested genotypes for common bunt resistance indicating a high level of variation in these genotypes. This variation was expected as the synthetic hybrid wheat was produced by crossing durum wheat (genome AB) with *Ae. Tauschii* (genome D). Common bunt occurs frequently in all wheat planting areas and causes a significant loss in yield quality and amount [28,34,35]. In addition, common bunt infection increases the susceptibility of winter wheat plants to winter injury [36]. Therefore, identifying new sources of genetic diversity for this trait is very helpful in breeding common bunt-resistant wheat genotypes that could be planted worldwide in organic and low-input farms. The high correlation between the resistance in Lincoln and Mead indicated that most of the tested genotypes had the same or similar response to common bunt across the two locations. However, a significant interaction between the location and genotype (G×L) was found. This significant G×L interaction was due to the different responses of a few genotypes to common bunt across the two locations. The high broad-sense heritability value indicated that common bunt resistance is a highly heritable trait and selection for this trait should be successful. High heritability for common bunt resistance was reported previously [28,37].

### 4.2. Expected Resistance Genes in the Studied Synthetic Wheat Genotypes

The worldwide set of the common bunt differential lines is very useful in identifying common bunt resistance genes in the germplasm of the tested genotypes. Using the differential lines for that purpose was performed previously [38,39,40,41]. These differential lines are available in the Department of Agricultural Research Service, National Small Grains Collection (NSGC) in Aberdeen, ID, and could be easily separated from each other, which makes them globally important [40]. They are very helpful in identifying the resistant genes as they are isolines carrying only one resistant gene [39,40,42]. Based on the results of the differential lines, *Bt6*, *Bt7*, *Bt9*, *Bt10*, *Bt11*, *Bt12*, *Bt13*, *Bt15*, and *Btp* are effective resistance genes to the Nebraska common bunt race. However, previous studies mentioned that *Bt5*, *Bt10*, *Bt11*, *Bt12*, and *Btp* were effective resistant genes to the common bunt race in Turkey [16]. The tested genotypes in this study were selected based on their resistance to the common bunt race in Turkey using the common race of the pathogen collected from Turkish fields (data not shown). Based on our study and previous studies, we predicted the genes that could exist in the germplasm of the 25 synthetic wheat genotypes as follows: (1) resistant genotypes to both common bunt races (Nebraska and Turkey) are expected to have *Bt10*, *Bt11*, *Bt12*, *Btp*, or other unknown resistant genes; (2) genotypes which are resistant to the Nebraska common bunt race and susceptible to the Turkish race could have *Bt6*, *Bt7*, *Bt9*, *Bt13*, *Bt15*, or other unknown genes; and (3) genotypes which have been found to resist the Turkish common bunt race and are susceptible to the Nebraska race could be carrying the *Bt5* gene or other unknown genes (Figure 4). Based on our results, the tested genotypes could be used as a possible genetic source of the different common bunt-resistant genes as was mentioned previously. Unfortunately, there are no useful DNA-based markers for the common bunt-resistant genes. Hence, we cannot confirm the presence of these genes in the elite 25 synthetic genotypes.

### 4.3. Genetic Variation in Some Agronomic Traits under Common Bunt Infection

The absence of significant differences among the tested genotypes for chlorophyll content and heading date under common bunt infection indicates that a common bunt infection does not affect these two traits. The presence of highly significant differences in seedling vigor and plant height among the tested genotypes indicates that the infection affected these two traits (Table 3). However, no correlation was found between the percentage of infected heads and seedling vigor or plant height. Based on our results, common bunt resistance seems to be an independent trait. Previous studies found no significant correlation between common bunt resistance and plant height in their tested populations, which confirmed our results [27,28]. Moreover, a previous study confirmed that plant height is controlled by a different genetic system than common bunt resistance based on a genome-wide association study, which provides more evidence that the two traits are independent of each other [27,28].

## 5. Conclusions

In conclusion, high genetic variation between the tested synthetic hybrid wheat genotypes was found. This variation showed that some of these genotypes could be used to improve wheat resistance to common bunt. Differential lines are very useful to identify the pathogenicity of disease and suggest the resistance genes that exist in the tested germplasm. Based on the differential lines results, the tested genotypes contain different resistance genes, which make them a possible source for common bunt resistance in Turkey and Nebraska. The identified resistant genotypes could be used as parents in the wheat breeding program, especially because they were selected based on agronomic traits, a high grain yield, and resistance to different biotic and abiotic stresses under Turkish conditions.

## Figures and Tables

**Figure 1 plants-12-00002-f001:**
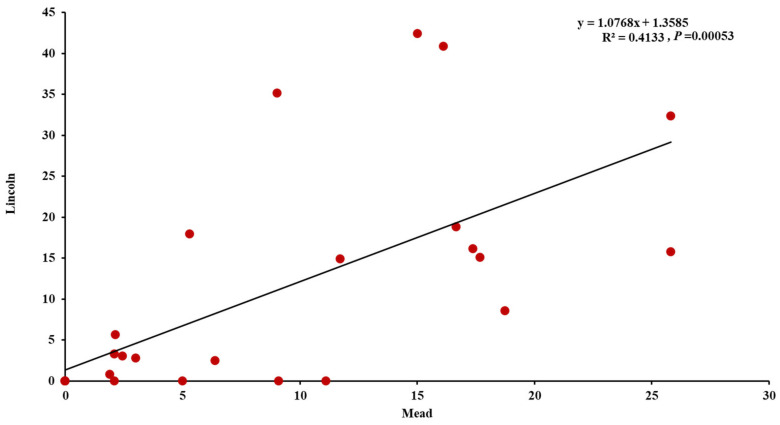
Correlation between percentage of infected heads under Lincoln and Mead conditions in the tested 25 wheat synthetic genotypes.

**Figure 2 plants-12-00002-f002:**
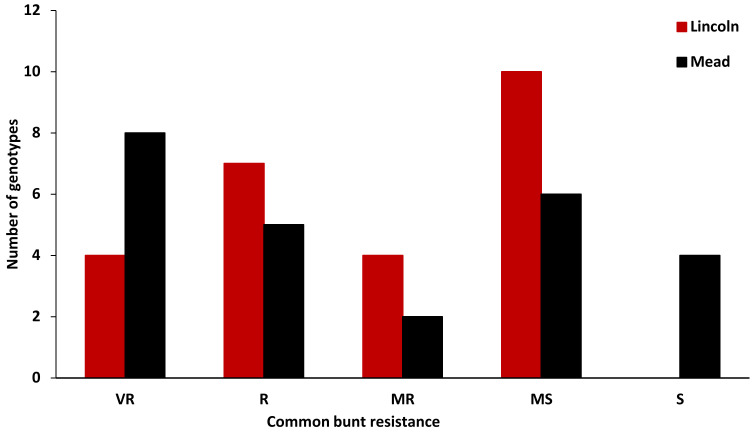
Number of genotypes representing different degrees of common bunt resistance at Lincoln (red columns) and Mead (black columns).

**Figure 3 plants-12-00002-f003:**
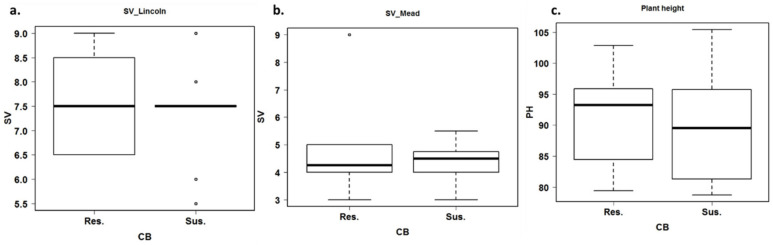
Box plots represent the differences in seedling vigor and plant height between common bunt resistance and susceptible genotypes. Seedlings of the resistant and susceptible genotypes were growing strongly at Lincoln with an average seedling vigor of 7.5 (**a**), and they were weak at Mead with an average of 4 (**b**). The plant height of the resistant and susceptible genotypes was almost the same with an average of 94 and 90 cm, respectively, based on the average of the two locations (**c**). Unlike seedling vigor, plant height did not differ significantly between locations, so the data from Mead and Lincoln were combined to perform the box plot.

**Figure 4 plants-12-00002-f004:**
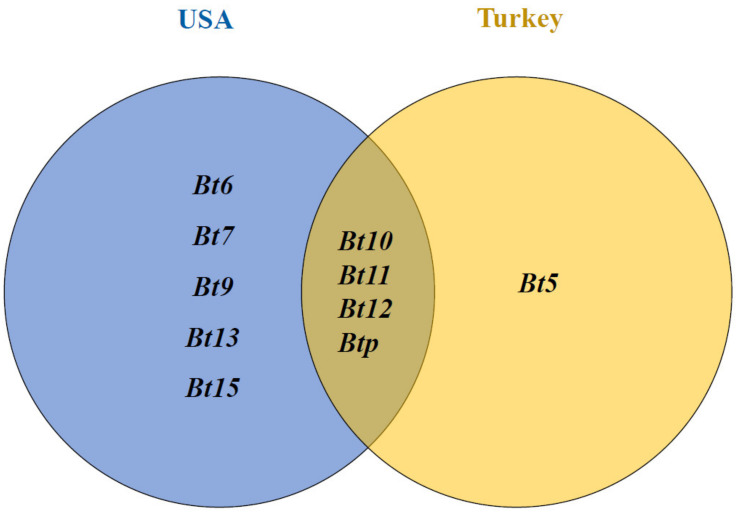
Venn diagram represents resistant genes for common bunt race/s in Nebraska (blue circle), Turkey (yellow circle), and both locations.

**Table 1 plants-12-00002-t001:** List of common bunt differential lines used in the current study, their PI number, and resistance genes.

Wheat Lines	Resistance Gene	CI or PI Number	% of Infected Heads with Common Bunt
Red Bobs	None	CI 6255	73.40
Heines VII	None	PI 209794	14.40
Sel 2092	*Bt1*	PI 554101	10.90
Sel 1102	*Bt2*	PI 554097	16.67
Ridit	*Bt3*	CI 6703	10.00
Rio	*Bt6*	CI 10061	0.00
Sel 50077	*Bt7*	PI 554100	3.80
PI 173438/Elgin	*Bt8*	PI 554120	12.50
Elgin/PI 178383	*Bt9*	PI 554099	0.00
Elgin/PI 178383	*Bt10*	PI 554118	1.20
Elgin/PI 166910	*Bt11*	PI 554119	0.00
PI 119333	*Bt12*	PI 119333	0.00
Thule III	*Bt13*	PI 181463	0.00
Doubbi	*Bt14*	CI 13711	33.3
Carleton	*Bt15*	CI 12064	0.00
PI 173437	*Btp*	PI 173437	0.00

**Table 2 plants-12-00002-t002:** List of synthetic hybrid wheat genotypes and their pedigrees used in the current study.

Genotype Number	Variety	Cross ID
**Checks**		
Gen. 1	KARAHAN	
Gen. 2	GEREK	
**Synthetics genotypes**		
Gen. 3 and 4	AISBERG/AE.SQUARROSA(369)	C04GH3
Gen. 5 and 6	AISBERG/AE.SQUARROSA(511)	C04GH5
Gen. 7	UKR-OD 1530.94/AE.SQUARROSA(310)	C04GH68
Gen. 8 and 9	UKR-OD952.92/AE.SQUARROSA(1031)	C04GH61
Gen. 10 and 11	UKR-OD 1530.94/AE.SQUARROSA(458)	C04GH74
**CYMMIT winter synthetics × modern varieties**		
Gen. 12 and 13	AISBERG/AE.SQUARROSA(369)//DEMIR	TCI091254
Gen. 14 and 15	LEUC 84693/AE.SQUARROSA(310)//ADYR	TCI091259
Gen. 16	LEUC 84693/AE.SQUARROSA(1026)//GEREK79	
Gen. 17 and 18	UKR-OD 952.92/AE.SQUARROSA(409)//SONMEZ	TCI091266
Gen. 19 and 20	UKR-OD 1530.94/AE.SQUARROSA(446)//KATIA1	TCI091274
Gen. 21, 22 and 23	UKR-OD 1530.94/AE.SQUARROSA(311)//EKIZ	
Gen. 24 and 25	UKR-OD 1530.94/AE.SQUARROSA(312)//BAGCI2002	TCI091272

**Table 3 plants-12-00002-t003:** Analysis of variance and broad sense heritability of common bunt resistance and studied agronomic traits (plant height and seedling vigor) in the two tested locations (Mead and Lincoln).

Source of Variance	Common Bunt Resistance		Seedling Vigor		Plant Height	
	d.f.	Mean Squares	d.f	Mean Squares	d.f	Mean Squares
Location (L) Replicate (R) Genotypes (G) G×L Error	1 1 24 24 40	109.80 141.23 394.87 ** 110.76 * 54.78	1 1 24 24 44	220.09 ** 3.16 2.76 ** 1.71 * 0.87	1 1 24 24 47	18.65 728.33 ** 280.17 ** 34.24 101.08
Broad-sense Heritability	0.86		0.64		0.69	

* *p* < 0.05; ** *p* < 0.01.

## Data Availability

Data is available with the corresponding author based on request.

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
