# Peer review of "Genetic Variation in Common Bunt Resistance in Synthetic Hexaploid Wheat"

_plants, 2022, doi:10.3390/plants12010002_

Round 1
Reviewer 1 Report
This is a very good paper and I enjoyed reading it. This paper can be accepted after major revision.
Please make all tables and figures self-explanatory
Please make statistical analysis in Figure 2.
Please make statistical analysis in Figure 3.
Please add 2022 references in the text and in the list.
There are many typos in the paper. Please correct it.
Many scientific names were written in NON italic format in both the text and the references. Please correct it.
I must revise the paper again.
Author Response
Dear Dr. Reviewer,
Thank you for your efforts and time in reviewing our manuscript. Please find a point-by-point response to your comments in the following lines.
Comments and Suggestions for Authors
This is a very good paper and I enjoyed reading it. This paper can be accepted after major revision.
Thank you for your nice words and great efforts.
Please make all tables and figures self-explanatory.
Titles of the figures and tables were adjusted accordingly.
Please make statistical analysis in Figure 2.
There was a mistake in this figure, and it was corrected.
Please make statistical analysis in Figure 3.
Not sure that I got your point here. How could we add statistical analysis for box plots?
Please add 2022 references in the text and in the list.
2022 Reference was added
There are many typos in the paper. Please correct it.
The manuscript was revised again carefully. Typos were corrected.
Many scientific names were written in NON italic format in both the text and the references. Please correct it.
They were corrected.
I must revise the paper again.
Reviewer 2 Report
The article presents the results of evaluating a set of Turkish synthetic wheat lines for resistance to Nebraska common bunt field population, and their performance in the two different environmental conditions. The research if original. The experimental design is appropriate. The discussion corresponds to the experimental data. In the whole, the conclusions are generally supported by the experimental data. The authors have supposed that the genotypes resistance to both Turkish and Nebraska common bunt Bt10, Bt11, Bt12, Btp, or other unknown resistance 271 genes
Comments.
1. The absence of the tables 1, 2 and 3, to which the authors refer substantially complicates understanding the results. The figures and discussion partly compensate the absence of such an important part. It is necessary to add the tables. Hope the tables are exist and this is only unfortunate omission.
2. The Introduction section can be improved by including the information on disease symptoms. A brief information on the known races of the pathogen, their distribution is also desirable.
3. The figure legends should be more informative. Please improve the legend to Figure 2. What does it mean “the degree of common bunt resistance”? What do mean the numbers on the axis Y?
4. Please the confidence level on the Figure 1.
5. Please give the authors names for Tilletia caries and T. foetida (line).
Author Response
Dear Dr. Reviewer,
We would like to thank you for your comments and efforts in reviewing our manuscript. Please find a point-by-point response to your comments.
Comments and Suggestions for Authors
The article presents the results of evaluating a set of Turkish synthetic wheat lines for resistance to Nebraska common bunt field population, and their performance in the two different environmental conditions. The research if original. The experimental design is appropriate. The discussion corresponds to the experimental data. In the whole, the conclusions are generally supported by the experimental data. The authors have supposed that the genotypes resistance to both Turkish and Nebraska common bunt Bt10, Bt11, Bt12, Btp, or other unknown resistance genes
Comments.
- The absence of the tables 1, 2 and 3, to which the authors refer substantially complicates understanding the results. The figures and discussion partly compensate the absence of such an important part. It is necessary to add the tables. Hope the tables are exist and this is only unfortunate omission.
The tables were included in the first submitted version of our manuscript. I believe that there was some problem/mistake by the journal may be. Please find the tables in the revised version.
- The Introduction section can be improved by including the information on disease symptoms. A brief information on the known races of the pathogen, their distribution is also desirable.
Thank you for your comment. A description of the disease symptoms was added.
- The figure legends should be more informative. Please improve the legend to Figure 2. What does it mean “the degree of common bunt resistance”? What do mean the numbers on the axis Y?
We apologize for this mistake. For some reason, this figure was replaced with other wrong one. Please find the right figure in the revised version. Figures legends were adjusted.
- Please the confidence level on the Figure 1.
It was added.
- Please give the authors names for Tilletia caries and foetida (line).
Names was added.